# Cereal Production Trends under Climate Change: Impacts and Adaptation Strategies in Southern Africa

**Luxon Nhamo** [1,*] **, Greenwell Matchaya** [1] **, Tafadzwanashe Mabhaudhi** [2] **,
Sibusiso Nhlengethwa** [1] **, Charles Nhemachena** [1] **and Sylvester Mpandeli** [3,4]

1   International Water Management Institute (IWMI-SA), 141 Cresswell St, Weavind Park,
    Silverton 0184, Pretoria, South Africa; G.Matchaya@cgiar.org (G.M.);
    S.Nhlengethwa@cgiar.org (S.N.); C.Nhemachena@cgiar.org (C.N.)
2   Centre for Transformative Agricultural and Food Systems, School of Agricultural, Earth and Environmental
    Sciences, University of KwaZulu-Natal, P/Bag X01, Scottsville 3209, Pietermaritzburg, South Africa;
    mabhaudhi@ukzn.ac.za
3   Water Research Commission, 4 Daventry Street, Lynnwood Manor, Pretoria 0081, South Africa;
    sylvesterm@wrc.org.za
4   School of Environmental Sciences, University of Venda, Private Bag X 5050, Thohoyandou 0950, South Africa
*   Correspondence: l.nhamo@cgiar.org; Tel.: +27-12-845-9100

**Abstract:** The increasing frequency and intensity of droughts and floods, coupled with increasing temperatures and declining rainfall totals, are exacerbating existing vulnerabilities in southern Africa. Agriculture is the most affected sector as 95% of cultivated area is rainfed. This review addressed trends in moisture stress and the impacts on crop production, highlighting adaptation possible strategies to ensure food security in southern Africa. Notable changes in rainfall patterns and deficiencies in soil moisture are estimated and discussed, as well as the impact of rainfall variability on crop production and proposed adaptation strategies in agriculture. Climate moisture index (CMI) was used to assess aridity levels. Southern Africa is described as a climate hotspot due to increasing aridity, low adaptive capacity, underdevelopment and marginalisation. Although crop yields have been increasing due to increases in irrigated area and use of improved seed varieties, they have not been able to meet the food requirements of a growing population, compromising regional food security targets. Most countries in the region depend on international aid to supplement yield deficits. The recurrence of droughts caused by the El Niño Southern Oscillation (ENSO) continue devastating the region, affecting livelihoods, economies and the environment. An example is the 2015/2016 ENSO drll for international aid to feed about 40 million people. In spite of the water scarcity challenges, cereal production continues to increase steadily due to increased investment in irrigated agriculture and improved crop varieties. Given the current and future vulnerability of the agriculture sector in southern Africa, proactive adaptation interventions are important to help farming communities develop resilient systems to adapt to the changes and variability in climate and other stressors.

**Keywords:** cereal production; climate change; adaptation; resilience; water scarcity; agriculture

---

## 1. Introduction

Understanding moisture trends (both soil moisture and rainfall) is critical in projecting future crop production under changing climatic conditions [1,2]. Previous studies have shown that climate variability and change will affect current and future farming and food systems as environments are substantially modified due to shifts in seasons [3–6]. For example, in southern Africa, variability and climate change and variability and other factors are adversely affecting the agriculture sector and the

capacity of the region to meet the food requirements to feed its growing population, mainly due to water deficits caused by increased demand from competing sectors as well as increased frequency and intensity of droughts [7,8]. Efforts to manage the stresses caused by the growing demand for food and water are hindered by a range of challenges such as increasing temperatures, changing rainfall patterns and rising sea levels, as well as land and water degradation [3,4,9]. Faced with such challenges, there is need to shift focus towards building resilience, strengthening adaptive capacity and build resilient food systems that are capable to progressively mitigate climate change impacts through the provision of innovative technologies, practices, systems and policies [6,9]. The knowledge of available adaptation options and their likely benefits or costs is highly linked to the knowledge of climatic trends and projections [10].

The sub-Saharan Africa (SSA) region is expected to be affected the most by moisture deficits as changes in rainfall patterns are resulting in changes in the area suitable for growing many crops [11–13]. The situation is worsened by reliance on rainfed agriculture, which may result in total crop failure if there is drought. For example, in the Southern African Development Community (SADC) only 6.6% of cultivated area is equipped for irrigation, which is a very small percentage of the irrigation potential of the region [14]. The demand for water has trebled since the 1950s, yet the availability of freshwater resources has been declining [15]. Although the impacts of climate change on food production are evident throughout the world, geographically they are unevenly distributed and the extent of the impacts vary from place to place, with losses of productive land felt mostly in arid and semi-arid regions [16]. The SADC region faces high risk of losing a greater proportion of its productive land due to climate change as seventy five percent of its area is arid or semi-arid [12].

The impacts of climate change on water and agriculture in southern Africa can be catastrophic as agriculture is the main sector, providing approximately 17% of regional GDP (increasing to above 28% when middle income countries are excluded) and contributing about 13% of the total export value and 60% of the region's population depend on the sector for their livelihood [14,17,18]. Therefore, climate change and its impacts on these climate sensitive sectors of water resources and agriculture are vital for the very survival of the region and its people. Temperatures have been increasing since the beginning of the 21st century [13,19], resulting in variation in the cropping season and affecting livestock breeds that were used to the normal temperatures and rainfall of the region. Trend analysis of temperatures across southern Africa indicate an increasing annual minimum and maximum temperatures at an average rate of 0.057 °C per decade and 0.046 °C per decade, respectively between 1901 and 2009 [13]. The Intergovernmental Panel on Climate Change (IPCC) acknowledges that developing countries, especially SSA, are extremely vulnerable to climate change impacts, as they lack the resources to adapt [20].

Impacts of changes in rainfall and temperatures patterns are already evident in southern Africa and these include: (a) high temperatures, which are causing increased evaporation and increasing water scarcity (water availability is expected to have halved by as early as 2025) [21,22]; (b) higher temperatures are causing the spread of pests and diseases, for example, the increasing frequency in armyworm and locusts invasion [20,23]; (c) capacity of reservoirs is diminishing due to the net effect of increased temperatures and evaporation; (d) increased frequency and intensity of droughts and floods [19,24]; and (e) increasing water pollution and a decreasing in water quality as a result of erosion and flash floods (these result in increased presence of sediments and nutrients in water bodies) [13].

This study examines trends in moisture variability in the Southern African Development Community (SADC) over the years, so as to make projections of future crop production. The review provides a detailed assessment of the region's water and food security challenges as they are related to climate change. The study further assesses the impact of the changes in rainfall patterns and variability, increasing aridity and variations in seasons on cereal yields in the region. Adaptation strategies that the region can adopt in order to have resilient communities are discussed. The aim was to provide evidence to policy and decision-making on cereal production as impacted by climate change in the region, as the region moves towards a transformative adaptation in the agricultural sector.

The SADC is an economic grouping of 16 countries: Angola, Botswana, Comoros, Democratic Republic of Congo, Lesotho, Madagascar, Malawi, Mauritius, Mozambique, Namibia, Seychelles, South Africa, Swaziland, Tanzania, Zambia and Zimbabwe (Figure 1). Of the total area of the region, which is 986,246,000 ha, only 6.11% is cultivated. The worsening water scarcity challenges are evidenced by the huge percentage of the regional area, which is arid (75%), meaning that only twenty-five percent is humid. The highly variable and unevenly distributed rainfall oscillates between 100 and 2000 mm/year [19]. Figure 1 is a locational map of the SADC region in Africa, also showing the unevenness in the distribution of annual rainfall.

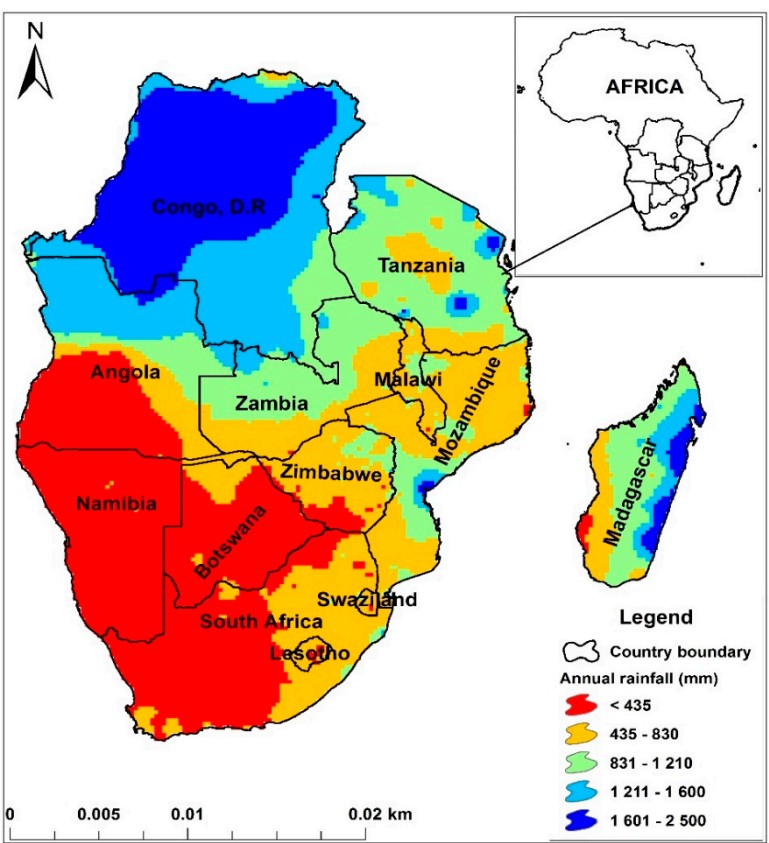

**Figure 1.** Locational map of Southern African Development Community (SADC) countries in Africa also showing annual rainfall distribution. Source: Developed by authors.

*Methodological Framework*

Climate change has been widely studied in southern Africa, showing historical trends of climatic variables over time as well as projected future scenarios and impacts [13,25–28]. Climate models agree that sub-Saharan Africa (SSA) will be greatly affected by climate change and variability due to underdeveloped, lack of resources to adapt and poor institutional and legal frameworks [1,13]. Whilst the anticipated climate change impacts in the region are well studied, little has been done on the impacts of the decreasing rainfall on cereal production. This review consulted literature on climate change trends in southern Africa, focusing on rainfall changes over time and the impacts on agriculture. The review first describes the agricultural potential of the region, followed by a discussion on the changes in moistures trends and patterns over time. The review further assesses the impact of the reductions in rainfall totals on cereal production. The purpose was to establish the relationship between the decreasing rainfall and cereal production. The review then highlights the climate change impacts in agriculture in general and proposes adaptation strategies that the region can adopt in order to have a resilient agricultural system. Available legal, policy and institutional arrangements at SADC region level are also discussed.

Observed rainfall data for a period of 47 years (1960–2007) obtained from 908 weather stations spread in the region was used to assess rainfall trends and aridity levels. Crop yield data for the same period was obtained from the Regional Strategic Analysis and Knowledge Support System (ReSAKSS) (www.resakss.org). Figure 2 shows the methodological flowchart followed in this review. The model depicts the linkages between policies and institutions, scenario planning and food security under climate change. Institutions and polices are framed around scenario planning to determine adaptation strategies for food security [29]. Policies and institutions provide the political will to exploit the agricultural potential under climate change.

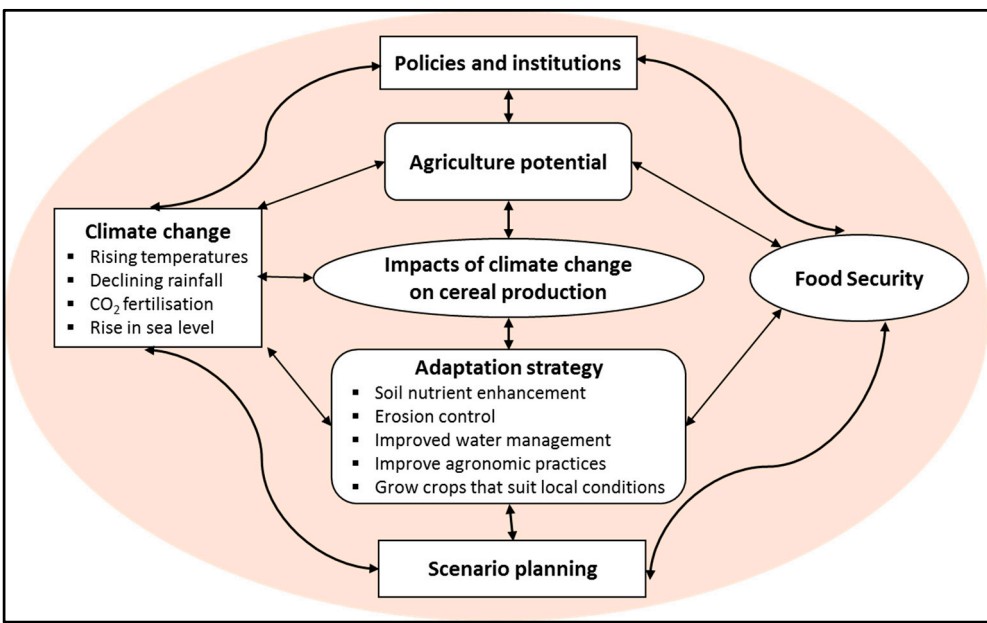

**Figure 2.** Methodological flowchart for climate change, cereal yield and food security. Source: Developed by authors.

## 2. Agriculture Potential of the SADC Region

The majority of the populace (60%) in southern Africa live in rural areas depending on natural resources for their livelihoods. Smallholder or small-scale farming is the main economic activity in rural areas, relying on seasonal rainfall as the main supply of water for crops and using traditional methods of farming [30]. Small-scale farming dominate the agriculture sector in terms of agricultural land coverage, cultivating about 80% of the cultivated land and contributing about 90% of agriculture produce [30]. Dependence on rainfed agriculture and the use of traditional methods of farming exacerbates the vulnerability of the smallholder farmers in particular and the whole region at large, to the vagaries of climate change, because once there is a drought there will be total crop failure [1]. These are some of the causes of the endemic food deficits in the region. Besides the huge investments in agriculture and the evidence of increased crop yield, agriculture continues struggling to meet the growing food requirements of an increasing population [1]. The level of food insecurity is already very high, particularly in rural settlements and many countries depend on international food aid to supplement for the yield deficits [1,31].

However, the SADC region is endowed with vast but underutilised agricultural land. Cultivated land accounts for only 6.11% of the total surface area of the region [32]. Although climate change could reduce the land suitable for agriculture, particularly rainfed agricultural land, irrigation using groundwater could revitalise such areas. Land with irrigation potential is about 20 million ha, yet only 3.9 million ha is actually irrigated, which accounts for about 6.6% of the total cultivated area. Although the irrigated area occupies only 6.6% of cultivated land, productivity from irrigated agriculture is three times bigger than productivity from rainfed agriculture [33]. There is potential of untapped

groundwater resources that can be used to supplement water deficits during periods of drought and during the dry winter season [1,34]. This is very crucial for a region where agriculture is the major driver of economic growth. As a result of the importance of agriculture in the region, there are huge agricultural investment plans aimed at ensuring food security and economic growth [35,36].

The success of these huge investments in agriculture is underpinned by adopting the water-energy-food (WEF) nexus as it provides evidence to policy and decision-making regarding resource utilisation and development, otherwise the investments will just transfer problems from agriculture to energy and water sectors [37]. In a WEF nexus perspective, developmental decisions are evidence based and ensures that targets are systematically achieved across sectors [38]. For example, it is suggested that increasing the area equipped with irrigation could provide opportunities for farmers to sustainably increase yield and address food insecurity and the Comprehensive Africa Agriculture Development Programme (CAADP) aims to increase irrigated area by 100% by 2025 from the baseline value of the year 2000; at an estimated cost of US$37 billion while infrastructure operation and maintenance required a further US$31 billion [35]. Whilst these strategies are noble, the question that immediately arises is, "where is the water to achieve these targets?" That is where the WEF nexus comes in handy, directing decision-making by evaluating planned interventions and establishing cross-sectoral interlinkages and interdependences [38]. The current sectoral approach in resource development and utilisation has become unsustainable as it causes imbalances in resource distribution and creating false shortages [39]. The WEF nexus ensures water, energy and food security and the attainment of Sustainable Development Goals and it can be linked to scenario planning [38,40]. About eight Sustainable Development Goals (SDGs) out of seventeen are directly or indirectly linked to agriculture, water and energy sectors highlighting the importance of the performance of these sectors to the attainment of inclusive economic development and transformation [39]. As global focus shifts towards the implementation and attainment of the 2030 Global Agenda of the SDGs, there is need to monitor and evaluate performance and progress of the Goals [38,39]. While regional targets on increasing the area under irrigation may contributes towards achieving Goal 2 of the SDGs, on zero hunger, knowledge of current trends in the agriculture sector as impacted by climate change is important to provide evidence to decision making [39]. The WEF nexus is important in assessing resource utilisation and development as well as monitoring SDGs, particularly Goals 2, 6 and 7 [38].

## 3. Changes in Moisture Patterns in the SADC Region

Climate change is acknowledged as the single most devastating challenge currently facing humankind, as its impacts are among the most urgent issues that decision makers have to deal with [1,25]. Greenhouse gas (GHG) emissions resulting from human activity are expected to contribute to the increasing global average temperatures by 2–5.4 °C by 2100 [41]. Global warming has already modified climate regimes of southern African by controlling the El Niño Southern Oscillation (ENSO) in the Pacific Ocean [42]. The ENSO affects the position of the Inter-tropical Convergence Zone (ITCZ) over southern Africa during summer. The ITCZ is a belt of low pressure that influences seasonal rainfall and temperature over southern Africa [43]. It is the driver of much variability of southern African rainfall. The ENSO has been impacting on climate regime of southern Africa, causing extreme weather events such as heatwaves, droughts and floods. The opposite of the El Niño is the La Niña that moves the ITCZ further south of the Equator, brings prolonged wet spells in southern Africa [43]. A recent study by the Council for Scientific and Industrial Research (CSIR) noted decreases in summer rainfall, shifts in seasons and increases in the frequency of heatwaves, droughts and floods in recent years [13]. The IPCC also projects that annual rainfall will be reduced by 20% by 2080 in southern Africa, a scenario that would worsen the vulnerabilities of the region [27]. The IPCC also notes that the region will experience an increase in the intensity and frequency of floods and droughts [27].

Negative changes in moisture regimes threaten the production of about 95% of agricultural land in southern Africa, as it is rainfed [44]. Projections indicate that southern Africa will be subjected to increased physical and/or economic water scarcity by as early as 2025 [27]. The IPCC estimates that

between 75–250 million people in Africa alone will be at high risk of increased water stress by as early as 2020 and the number could increase to between 350–600 million people by 2050 [27,28]. Reduced rainfall and increased temperatures are gradually reducing (a) the area suitable for agriculture, (b) the length of growing period and (c) yield potential [27,28]. By 2080, rainfall variability and longer dry spells would result in reduction of crop yields, rise in sea levels and coastal and low-lying areas would be affected by floods. Livestock breeds that are adapted to current climate regimes will greatly be affected and their populations reduced due to new strains and diseases.

There are already significant changes being experienced in agriculture, as well as other sectors such as water, energy, biodiversity and health, due to climate change and variability [45]. Besides causing low crop yields, depleting freshwater resources are also limiting hydropower generation, which is the main source of energy in the region [38]. Increasing temperatures and declining rainfall are adversely degrading ecosystems and thus affecting the ecosystems services they provide [5,9]. There are also health and social risks associated with climate change, such as the impacts on vulnerable children and the elderly, pregnant women, social marginalisation (associated in some areas with indigenous populations, poverty or migration status), among others [16,24,31]. Water-borne diseases and malaria are expected to increase as a result of climate change, apart from the anticipated new and emerging health issues related to heatwaves and other extreme climate events [24,31]. These challenges are derailing progress made so far in poverty alleviation, employment, housing, access to and provision of basic services, food security and provision of potable water [18,46].

Figure 3 represents the variations in the annual average rainfall per season (summer, autumn, winter and spring) from 1960 to 1996 in southern Africa. In the SADC region, the summer season (December to February) is wet as it receives rainfall but the winter season (June to August) is dry as it receives insignificant rainfall [13]. Autumn (March to May) and spring (September to November) are transitional seasons ushering into winter and summer, respectively. Rainfall shows high variability from 1960–1961 to 1988–1989 (Figure 3) but as from 1989–1990 and thereafter, the variability is associated by a continuous decrease in rainfall totals during the summer periods, signifying a shortened rainy season. Drought recurrence explains the decreasing rainfall and this has been impacting on agricultural productivity as also suggested by previous studies [47–49].

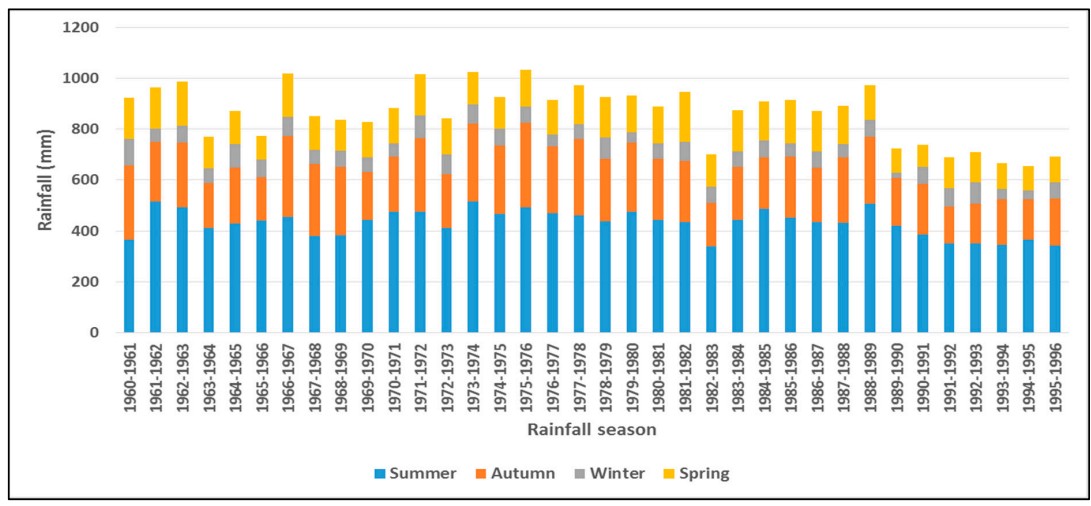

**Figure 3.** Temporal change in annual and seasonal rainfall in the SADC region from 1960 to 1996. Source: Developed by authors.

The increasing aridity of the SADC region is indicated in Figure 4, which shows changes in water scarcity from 1980 to 2007. The aridity or water scarcity over time was calculated through the climate moisture index (CMI) using Equations 1 and 2 [50]. When CMI values are negative or below 0, it indicates that potential evapotranspiration (PET) exceeds precipitation (P). According to Vörösmarty et al., there is a classification link between CMI values and climatic conditions (CMI < −0.6 = Arid; −0.6 < CMI <0 = Semi-arid; 0 > CMI < 0.25 = Sub-humid and CMI > 0.25 = Humid) [50]. The average CMI for the region was calculated at −0.80, qualifying the region to be arid and water scarce according to the scale provided by Vörösmarty [50]. The level of aridity in the region is also confirmed by previous studies that show an increasing level of aridity [51]. The increasing aridity is only worsening the vulnerability of the region that depends on agriculture. The CMI is effectively used in assessing aridity and water scarcity and is expressed as:

$$CMI = \frac{P}{PET} - 1, \text{ when } P < PET \tag{1}$$

$$CMI = 1 - \frac{P}{PET}, \text{ when } P \geq PET \tag{2}$$

where, PET is potential evapotranspiration and P is precipitation.

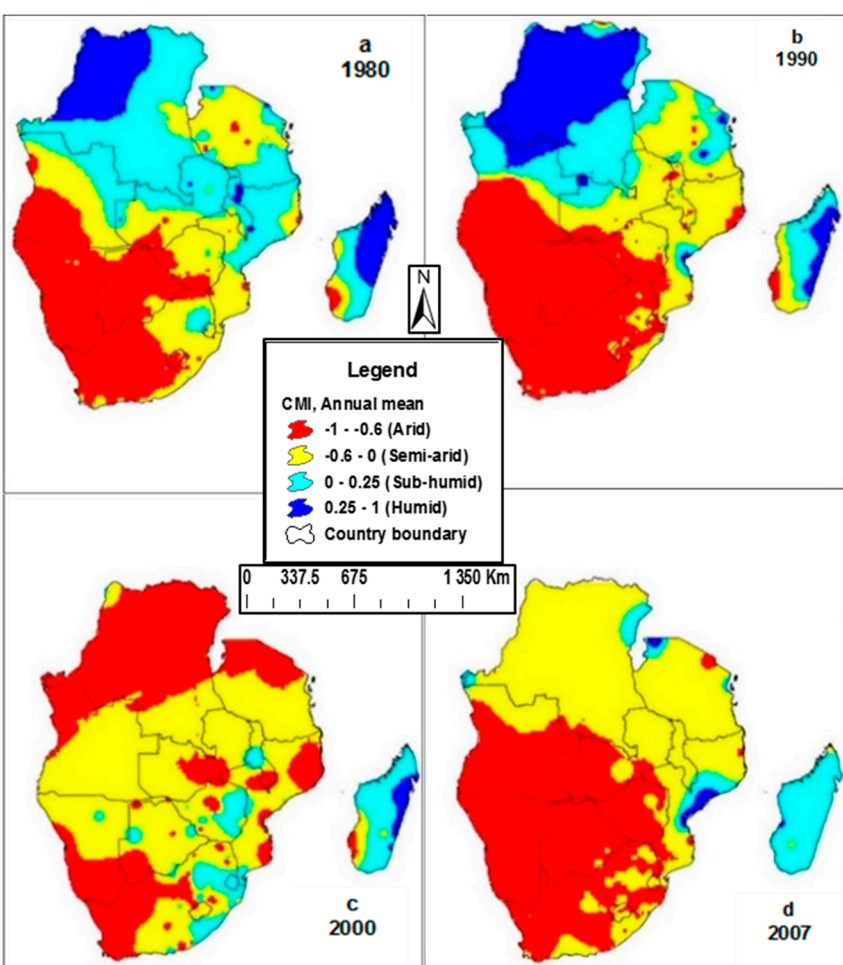

**Figure 4.** Spatio-temporal changes in water scarcity and aridity in the SADC region. Source: Developed by authors.

Although the Comprehensive Africa Agriculture Development Programme (CAADP), through the New Partnership for Africa's Development (NEPAD) in partnership with the United Nations' Food and Agriculture Organisation (FAO), intends to increase the irrigated area in Africa by 100% by 2025 from the baseline value of the year 2000 at an estimated cost of US$37 billion [35,36], the increasing drought recurrence and intensity, as well as the increasing aridity present the greatest obstacle for the region to meet the target. These are the greatest challenges that are impeding the region from meeting its agriculture targets of ensuring food security and poverty eradication [1,52].

## 4. Impacts of Rainfall Variability on Cereal Production

Moisture variations over southern Africa are adversely affecting crop yields due to water stress as shown by the increasing aridity levels and the shortened agriculture season (Figures 3 and 4 respectively). Dependence on rainfed agriculture, as well as other factors like poor infrastructure and lack of resources to adapt to the changing environment only increase the vulnerability of the region [1,53,54]. As a result, agricultural productivity in southern Africa is anticipated to decrease from 15% to 50% by 2080 due to climate change and variability [55]. Apart from food insecurity, other sectors like water and energy would also be greatly affected [56]. The largest proportions of vulnerable populations to these vagaries of climate change are invariably found in the rural areas where 60% of the population resides. Rural communities in southern Africa are characterised by chronic water, food and energy insecurity and malnourishment remain endemic [56].

According to the IPCC, SSA countries will suffer losses of between 2% and 7% in agricultural GDP (AgGDP) by 2100 [26,57]. These changes will occur at a time when population is projected to increase from 0.9 billion people in 2005 to about 2 billion by 2050 [26]. Although crop production is steadily increasing in the region, the anticipated losses in AgGDP, coupled with population growth mean that agriculture would not be able to feed the growing population if no action is taken to mitigate the challenge [28,57]. Maize production, the staple cereal of the region, is anticipated to suffer production reductions of between 12 and 40% by 2050 [58,59]. The IPCC reports anticipated losses of between 27–32% in the production of maize, sorghum, millet and groundnut for a warming of about 2 °C above pre-industrial levels by 2050 [27]. Estimates from previous studies also show that crop and fodder growing periods in southern Africa would shorten by an average of 20% by the mid-century, causing a 40% reduction in cereal yields and a decline in cereal biomass for livestock [27,59]. The empirical evidence of the impacts of climate change in southern Africa indicate extensive and dire consequences for agriculture if no mitigatory measures are put into practice.

Figure 5 presents the relationships between rainfall variability and cereal productivity in the SADC from 1960 to 2007. Cereal production steadily increased over the years from 14 million tonnes in 1960 to 26 million tonnes 2007, representing an increase of 85%. However, there were years in which yields would drastically drop due to droughts. Examples of extreme reductions in cereal yields occurred during the 1982/1983 and 1991/1992 agricultural season due to the droughts in those years [1]. The annual rainfall graph depicts a decreasing trend with high levels of variability. Rainfall is highly variable and the variability has been intensifying in recent past as rainfall totals decrease. This is ascertained from the coefficient of variation values of annual rainfall as the Mann-Kendall trend test (a statistical assessment of whether a set of data values is increasing or decreasing over time and whether the trend in either direction is statistically significant) shows a very significant decreasing trend of $\alpha < 0.01$ in rainfall totals over the years. In spite of the declining trend in rainfall totals, cereal productivity has been steadily increasing mainly due to increased investment in the agriculture sector and the increase in cultivated area [60]. The increase in yields could also be due to improved agronomic practices such as the use of improved seed varieties, fertilizer and pesticide application, capacity building and technology transfer. Extensification and intensification of irrigated agriculture is also contributing towards the increased cereal production [60]. However, it should be noted that South Africa is contributing a lot to the increasing crop yield as most countries always fail to meet their food requirements and generally depend on international aid to supplement their yield deficits as shown in

Table 1. Whilst 95% of the agriculture in the rest of other SADC countries is rainfed and vulnerable to climate change and variability, 30% of South Africa's agriculture produce comes from irrigation and it exports part of the excess produce to some countries in the region [33]. It should also be noted that population has been increasing during the same period, resulting increased demand for food, which resulted in constant shortfalls of cereals [61].

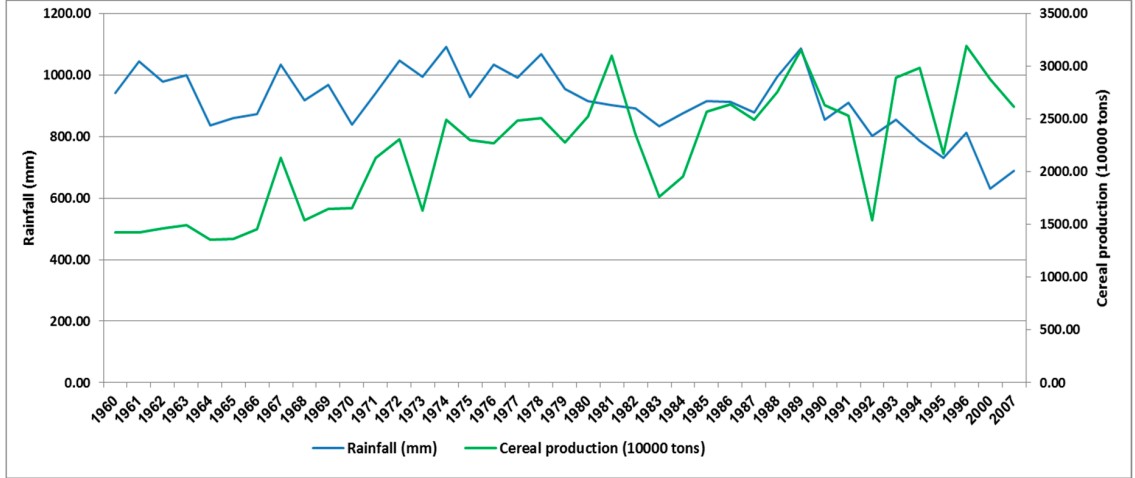

**Figure 5.** Relationship between rainfall variability and cereal production in SADC countries. Source: Developed by authors from Regional Strategic Analysis and Knowledge Support System (ReSAKSS) data (www.resakss.org).

Major cereals consumed in the SADC region include millet, rice, sorghum, barley, wheat and maize, maize being the staple food crop consumed in the region. Maize production has been going down in most countries in the region mainly due to extreme droughts and floods. Table 1 shows the deficits in maize production in most countries in the region since 2011, except Zambia, which shows minimal surplus. Maize is the staple food crop in southern Africa. Although the period covered includes the famous drought of 2015/2016, this has been generally the trend in the region. During this period even South Africa, the regional economic and food hub, had yield deficits and had to import food [1]. Therefore, maize production is susceptible to drought conditions and it is affecting food security in the region. The sustainability of maize production is essential for both national and regional economic development and improvement of rural livelihoods, yet yields have been going down.

**Table 1.** 2016 maize production deficit in SADC countries and the number of affected people.

| Country | 2011–2015 Average (1000 tons) | 2015 Maize Production (1000 tons) | 2016 Maize Production (1000 tons) | % Change 2015/2016 | No. of Affected People in 2016 |
|---|---|---|---|---|---|
| Angola | 1366 | 1878 | 1500 | −20 | 756,000 |
| Botswana | 21 | 4 | 1 | −75 | 1,100,000 |
| Lesotho | 74 | 79 | 25 | −68 | 709,000 |
| Madagascar | 393 | 350 | 300 | −14 | 1,400,000 |
| Malawi | 3583 | 2776 | 2369 | −15 | 6,500,000 |
| Mozambique | 1602 | 1357 | 1350 | −1 | 2,000,000 |
| Namibia | 61 | 38 | 46 | 21 | 729,000 |
| South Africa | 12,345 | 10,629 | 7733 | −27 | 14,300,000 |
| Swaziland | 89 | 82 | 33 | −60 | 638,000 |
| Zambia | 2894 | 2618 | 2873 | 10 | 976,000 |
| Zimbabwe | 1083 | 742 | 512 | −31 | 4,000,000 |

Source: FAO GIEWS, 2016 [62].

## 5. Adaptation Strategies in the Agriculture Sector and Policy Implications

The projected increase of more than 2 °C in temperature over the globe requires long-term adaptation policy strategies designed to reduce the impacts of climate change. Global warming poses the greatest challenge to policy and decision-makers in attempting to address the unpredictable changes in climate [63]. Because of the uncertainty in climate projections, decision-makers are opting to use scenario planning, whereby planning for future climate changes takes into consideration the limitations of climate projections, particularly uncertainty, in order for decision-makers to plan for future conditions outside observed trends [64,65]. Scenario planning uses the "what if, when and how" approach rather than a predictive one which has lots of uncertainty [66]. As a practical way of identifying a range of future conditions, scenario planning considers alternative response options [65]. The main advantage is that scenarios are not predictions and they do not assign likelihoods to particular future conditions but they broaden opportunities for decision-making to include a range of potential responses [65]. Thus, decision-makers have the flexibility to plan for an unpredictable future outside uncertainty. The main challenge is to design adaptation policies that would be flexible enough to address a variety of future scenarios [65]. Some scenario planning methods that are being used in climate change adaptation include the Shared Socioeconomic Pathways (SSPs) and the Representative Concentration Pathways (RCPs). Shared Socioeconomic Pathways and Representative Concentration Pathways are a set of pathways and frameworks developed by the climate change community to facilitate an integrated analysis of long-term and near-term modelling experiments for climate change to assess vulnerabilities and recommend adaptation and mitigation strategies [67–69].

The premise is that policies on climate change adaptation need to be aligned to governance capabilities such as (a) reflexivity, (b) resilience, (c) responsiveness and (d) revitalisation. Reflexivity is the capability to systematically and continuously deal with a variety of problem as they emerge; resilience is the ability to bounce back to the original basic state of function after a perturbation; responsiveness is the ability to deal with dynamic demands and expectations, and; revitalisation is the ability to reignite policies and ensure their continuous application [70]. Ideally, these approaches must be flexible to allow upscaling and downscaling, depending on and in response to the prevailing challenges at local and transboundary scales [71]. In addition, adaptive management, which allows for iterative decision, is needed to manage climate change risk and uncertainty.

An important consideration when developing adaptation strategies is to understand how sectoral and cross-sectoral climate change impacts affect the rural and urban poor [72]. As climate change is a crosscutting, adaptation strategies should be framed in the context of the WEF nexus, which ensures that they are evidence based, coordinated and coherent [37,56]. The SADC Secretariat produced a policy paper on climate change, which stresses on cross-sectoral approach to mitigate climate change impacts [73]. The policy paper highlights two key aspects for a future regional climate change programme; (a) to establish an implementation strategy and (b) to develop an action plan. Another important document on climate change adaptation at SADC regional level is the Climate Change Adaptation Strategy for the Water Sector, which stresses the need to enhance climate resilience through integrated and adapted water resources management at all levels (regional, river basin and local levels) [74]. The SADC Regional Agricultural Policy (RAP) [14] is another important policy document in climate change adaptation as it emphasises on enhancing agriculture productivity to meet regional food and water requirements, as well as attaining sustainable economic development objectives at a regional level [14]. In addition, the SADC's Regional Indicative Strategic Development Plan (RISDP) and the RAP propose to increase the irrigated area in Africa by 100% by 2025 from the baseline value of the year 2000 [35].

Table 2 provides some of the risks presented by climate change in the agriculture sector and the possible adaptation strategies. A key component of climate adaptation is building resilience, the capacity of a system to absorb disturbance without collapsing [75]. A resilient system withstands shocks and rebuilds itself when necessary. Adapting to the risks brought about by climate change are possible by adopting strategies such as autonomous adaptation (shifts in planting dates and cultivar

substitution) and embracing new technologies and transformational changes (climate-smart agriculture and livelihood diversification or change), as well as improving on trade policies and encouraging shifts in diets (Table 2).

**Table 2.** Climate change risks on agriculture and adaptation strategies.

| Climate Risk | Proposed Adaptation Strategy |
|---|---|
| Increasing climate variability, frequency and intensity of extreme weather events. | Access to climate information and services for improved decision-making. Early warning systems to mitigate the impact of extreme weather events such as drought and floods. |
| Decreased availability of freshwater resources in waterbodies due to increased open water evaporation. | Promoting water efficient irrigation technologies and climate smart agricultural practices that maximise water productivity, i.e., more crop per drop. Timing of crop production calendars to maximise irrigation water use in dams and reservoirs in order to mitigate losses to open surface evaporation. |
| Increased danger of water pollution and decreased water quality, which may cause water borne diseases. | Improve on agronomic practices and reduce nutrients from reaching water-bodies. |
| Decrease in crop productivity due to increasing rainfall variability, decreasing rainfall totals and shifting seasons. | Promote water efficient irrigation technologies and strategies such as drip irrigation and deficit irrigation, respectively. Promoting the breeding of drought tolerant and water efficient crops suitable for low and variable rainfall environments. This would also include promotion of underutilised indigenous and traditional crops that have shown adaptation to these environments. Shifting seasons and season duration can be addressed by shifting planting dates from traditional dates as well as matching varieties to environments, e.g. where seasons are becoming shorter, the promotion of early maturing varieties would be suitable. Improving access to water in some areas will help to extend the length of the season and may facilitate all year cropping. |
| Increasing temperatures result in heat stress and the spread of pests and pathogens. | The breeding and promotion of drought and heat stress tolerant crops that are adapted to extreme weather conditions. This would also include promotion of underutilised indigenous and traditional crops that have shown adaptation to these environments. Cultivation of crops with resistance to pests and diseases as well as climate smart agricultural practices that promote greater crop diversity and promote biological control of pests and diseases. In some cases, adaptation may have to include controlled environment production (tunnels and greenhouses), especially for high value and sensitive crops. |
| Increased incidence of floods and flash floods as a result of sudden heavy downpours. This is also associated with erosion of the top soil and loss of carbon from top soils. | Develop and promote rainwater harvesting and conservation practices to capture and store the excess rainfall and save it for use during dry periods. This will also contribute to extending season length by increasing access to water. Cultivation of cover crops and promotion of climate smart agricultural practices that promote permanent ground cover to mitigate runoff. |

Agriculture is highly dependent on climate, making the sector highly vulnerable to climate change and variability. Although increases in temperature and carbon dioxide ($CO_2$) may result in the increase of some crop yields, certain conditions such as improving on nutrient levels, soil moisture and water availability must also be met [76,77]. Generally, climate change has the risk of modifying ecosystems and make it more difficult to grow crops in the same traditional ways and same environments as before [78]. As shown in Table 1 extreme temperatures and precipitation are anticipated to affect crop production and droughts are already causing total crop failure in areas without irrigation to

supplement crop water deficits during dry periods [1]. Weeds, pests and diseases can flourish under increased temperatures affecting both crops and livestock [54,79]. Although rising $CO_2$ may stimulate plant growth, research has also shown that it reduces the nutritional value of food crops as high concentrations $CO_2$ in the atmosphere reduces the amount of protein and other important minerals in most cereals and other plant species [80–82].

As climate change impacts are already evident in the agriculture sector, there are innovative tools that have been developed to identify appropriate adaptation options. A good example is the Climate Analogues, which emphasises on finding future agriculture today [83]. The tool, developed by the CGIAR Research Program on Climate Change, Agriculture and Food Security (CCAFS), identifies geographic areas whose current growing conditions present similar projected future climates and allows communities living in such areas to exchange ideas and learn on agriculture practices that work well in those future environments. This promotes an understanding of adaptation practices that can be adopted to local context to cope with anticipated shifts in growing conditions over time [83].

## 6. Conclusions

The evident and adverse climate change impacts on the agriculture sector in southern Africa that include reduced crop yields and new strains of pests and diseases, require evidence based adaptations policy frameworks that leads to the resiliency of the agricultural system. The region's growing aridity levels are exacerbating poverty, high unemployment levels, inequalities and vulnerabilities. Although cereal production has been steadily increasing over time, it has not been able to feed a growing population, as most countries in the region still rely on international aid to supplement yield deficits. This is besides the huge investments being directed towards the agriculture sector and the immense agricultural potential of southern Africa. The region needs to develop flexible adaption strategies and frameworks framed around scenario planning that would culminate in a resilient agricultural system by increasing knowledge exchange and shared best practices on detecting pets and diseases, as well as weather forecasting and drought monitoring, among others. Although these can be implemented at national and local levels, they are more effective at regional level as dealing with the challenges at regional levels reduces the risk of recurrence. Implementing these initiatives at regional could be very positive for a region that is moving towards integration. Agriculture development and adaptation to climate change should consider cross-sectoral approach of the water-energy-food (WEF) nexus, which systematically provides evidence to policy and decision-making. The advantage of the WEF nexus is that it is flexible and can be linked to scenario planning methods such as Shared Socioeconomic Pathways (SSPs) and the Representative Concentration Pathways (RCPs), as well as adaptation tools like the Climate Analogues.

**Author Contributions:** L.N and G.M designed the study concept, literature review, data collection and analysis. T.M., C.N., S.N and S.M. made data analysis, wrote parts of the manuscript and interpreted data.

**Funding:** This research was funded by the Bill & Melinda Gates Foundation (BMGF) and the United States Agency for International Development (USAID) through the International Food Policy Research Institute's (IFPRI), the Regional Strategic Analysis and Knowledge Support System (ReSAKSS-SA) programme.

**Acknowledgments:** The authors would like to thank the International Water Management Institute (IWMI) for supporting the ReSAKSS programme and for the support in writing this paper.

**Conflicts of Interest:** The authors declare no conflict of interest.

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
