# Peer review of "Cereal Production Trends under Climate Change: Impacts and Adaptation Strategies in Southern Africa"

_agriculture, doi:10.3390/agriculture9020030_

Round 1

Reviewer 1 Report

The authors provide an interesting review and discussion of the impacts of climate change on cereal productions in Sub-Saharan Africa as well as a discussion of a number of efficient adaptation strategies. This paper is rather well written and provides a comprehensive overview of the current situation and future impacts, and does a good job in synthesizing current knowledge and pointing out potential adaptation strategies. This is an important topic and a needed discussion paper, I first have a number of concerns and comments that I would like to see addressed. The main comments are:

Abstract: Since the paper also offers a review of both impacts and adaptation strategies, the abstract should reflect this. However, the current version of the abstract discusses only impacts, and not adaptation strategies. These should be discussed too in the abstract.

Introduction: It is not clear what the aim of this study really is (at least when reading the introduction). Please revise the paragraph L79-86 to state clearly what the objectives of this paper are and how it is structured. Moreover, since this paper is a review paper, at least one paragraph should explain how literature was screened and which studies were selected to contribute to the review.

Section 2: I find it hard to make sense out of the numbers provided in this section (particularly the second paragraph). Moreover, the connection with the SDG is interesting, but you should elaborate more on this. I believe that there is a number of interesting papers on this topic, which should be included there.

Section 3: This section is crucial to the paper, but lacks of references. Considering the large number of studies that have explored impacts of climate change on moisture patterns in SSA region, it seems that there is far more to say – and far more studies to include – than it is currently the case in the paper. Please provide an overview of the (1) up to date and (2) state of the art studies linked to this topic. The emphasis is also too much on the past variations (e.g. Figure 2 and 3) rather than on future impacts under changing climatic conditions – which should be the major focus.

Section 4 (on impacts): Again, this section and review of the impacts of rainfall variability on cereal production should be based on a larger number of studies. This section is too discursive and qualitative, and lacks of quantitative aspects. Future aspects – under climate change – should also be discussed.

Section 4 (on adaptation): Please rename this section “Section 5”. This section could be enhanced (particularly Table 2) with a larger number of adaptation options and strategies. These could also be more specific, as the current adaptation strategies are very broad and falls short in addressing the specific issues that you raised throughout the paper.

Conclusion: this section falls short in reflecting upon the paper and providing new insights and further research. I would like to see more discussion on what is needed and what are the different steps (both in terms of research and policy-making) that should be made in order to increase resilience of cereals production under changing climatic conditions in the SSA region. I would particularly like to see a paragraph on how social policies to reduce vulnerabilities could play an important role in increasing resilience and adaptive capacity of the local populations.

I also have a number of minor comments:

- L26: “growing” instead of “bourgeoning”.

- L33: Please define WEF prior to the use of the acronym.

- L38: “under changing climatic conditions” instead of “in an environment of climate change”.

- L41: In which region of the world? Worldwide? Africa? Sub-Saharan Africa? Please clarify.

- L42: Is water deficits caused by competing sectors or by climate change (or both)? Please clarify.

- L45: Is rising sea level an important climatic determinant when it comes to food security? If so, please provide references for this statement (currently, ref number 9 is too broad to support this statement).

- L56: What is a “small” impact? Please clarify (I also suggest avoiding using such fuzzy words).

- L61: “on water resources”.

- L65: Any uncertainty range for these numbers?

- Figure 1: It would be great to add more information in this Figure, as a basemap. For instance, the different types of climatic zones or the different types of cereals productions.

- L117: That is instead of “that’s”.

- L124-132: Is this paragraph really needed (or need to be that long)? This is quite technical and off-the topic. I would suggest reducing the length of this paragraph, while increasing the length of other paragraphs (based on a wider review of existing studies).

- L133: redundancy

- L143: what are these changes? Please specify and elaborate on this.

- L145: Please provide more references to support this crucial claim. The ref 42 is only about food security, and not about housing, employment, etc.

- Figure 2: As said, it would be great to also have a figure showing future impacts of climate change (e.g. under different emissions scenarios) on temporal change in annual and seasonal rainfall in the SADC regions. This way, readers can easily grasp the extent of the impacts, compared to historical conditions.

- L160: Please provide a scale of the CMI values, as many readers might not be aware of how the CMI is computed, what is its scale, and hence what -0.80 means (High? Medium? Low?).

- L163: what is huge? Please clarify (and I suggest avoiding the term ‘huge’ in any case, as it is highly subjective).

- Figure 3: since this is a review paper, I suggest including a figure from a different source, rather than using another figure from Nhamo et al.

- L180-184: Please give some numbers and references to support these statements.

- L185: Please clarify what is the Mann-Kendall trend and what a value <0.01 means. A number of readers might be unaware of this trend index.

- L194: Why is that? Please explain how different South-African agriculture is.

- Figure 4: what is ReSAKSS data? Please specify and define the acronym prior to its use.

- Table 1: should be more discussed in the text.

- L212: socioeconomic and climatic changes are two different things.

- L214: you could refer here to the new IPCC scenarios, i.e. socioeconomic scenarios (Shared socioeconomic pathways -- SSPs) and climate scenarios (Representative concentration pathways --RCPs).

- L215: please provide a few definitions for the terms that you use (reflexivity, responsiveness, and revitalization).

- L235: At the end of this section, you could also briefly discuss a few innovative tools that can help identify appropriate adaptation options. One of this tool is the climate twins approach (Rohat et al. 2017 https://link.springer.com/article/10.1007/s11027-016-9708-x ; Rohat et al. 2018 https://www.emeraldinsight.com/doi/full/10.1108/IJCCSM-05-2017-0108), which has been applied in Sub-Saharan Africa to enhance resilience and food security at the local scale (see report from CGIAR https://ccafs.cgiar.org/publications/climate-analogues-finding-tomorrow%E2%80%99s-agriculture-today#.XCycCFxKiUk).

- L247: Conclusion should not finish on such a negative side, but instead should reflect and discuss the way forward, and opportunities, and potential solutions to increase resilience.

Author Response

Thank you for the time to review the manuscript. We have extensively revised the manuscript according to comments received, and changes are shown in blue.

Responses to comments are given in blue

The authors provide an interesting review and discussion of the impacts of climate change on cereal productions in Sub-Saharan Africa as well as a discussion of a number of efficient adaptation strategies. This paper is rather well written and provides a comprehensive overview of the current situation and future impacts, and does a good job in synthesizing current knowledge and pointing out potential adaptation strategies. This is an important topic and a needed discussion paper, I first have a number of concerns and comments that I would like to see addressed. The main comments are:

Thank you for the valuable comments, which we used to improve the manuscript. The paper has been extensively revised.

Abstract: Since the paper also offers a review of both impacts and adaptation strategies, the abstract should reflect this. However, the current version of the abstract discusses only impacts, and not adaptation strategies. These should be discussed too in the abstract.

We added a sentence highlighting that the review proposes some adaptation strategies in the agriculture sector. We also added a statement on adaptation in the conclusions

Introduction: It is not clear what the aim of this study really is (at least when reading the introduction). Please revise the paragraph L79-86 to state clearly what the objectives of this paper are and how it is structured. Moreover, since this paper is a review paper, at least one paragraph should explain how literature was screened and which studies were selected to contribute to the review.

We added the study objectives in the highlighted text of the last paragraph of the introduction. We also added section 1.1 on the methodological framework where we detail how the study was done, including literature review and a flowchart.

Section 2: I find it hard to make sense out of the numbers provided in this section (particularly the second paragraph). Moreover, the connection with the SDG is interesting, but you should elaborate more on this. I believe that there is a number of interesting papers on this topic, which should be included there.

We have improved the wording of the 2nd paragraph of section 2 to better explain the used statistical information. We have also further clarified the connection between agriculture monitoring, WEF nexus and SDG in the last part of section 2

Section 3: This section is crucial to the paper, but lacks of references. Considering the large number of studies that have explored impacts of climate change on moisture patterns in SSA region, it seems that there is far more to say – and far more studies to include – than it is currently the case in the paper. Please provide an overview of the (1) up to date and (2) state of the art studies linked to this topic. The emphasis is also too much on the past variations (e.g. Figure 2 and 3) rather than on future impacts under changing climatic conditions – which should be the major focus.

We have added information and references on more recent data on Section 3. The added references include projections by the IPCC on rainfall of the region. The added information (highlighted in blue) provides an overview of current and future rainfall situation in southern Africa. We lacked enough information on most recent observed data on Figure 2 and 3 as data was unavailable or costly. We used the available dates in order to have the same the data with crop yield.

Section 4 (on impacts): Again, this section and review of the impacts of rainfall variability on cereal production should be based on a larger number of studies. This section is too discursive and qualitative, and lacks of quantitative aspects. Future aspects – under climate change – should also be discussed.

We have also added a paragraph information in Section 4 highlighting the future impacts of climate change in agriculture

Section 4 (on adaptation): Please rename this section “Section 5”. This section could be enhanced (particularly Table 2) with a larger number of adaptation options and strategies. These could also be more specific, as the current adaptation strategies are very broad and falls short in addressing the specific issues that you raised throughout the paper.

We have extensively revised Section 5 by adding scenario planning as a better means of climate change adaptation in the agriculture sector. We have also added more strategies to Table 2.

Conclusion: this section falls short in reflecting upon the paper and providing new insights and further research. I would like to see more discussion on what is needed and what are the different steps (both in terms of research and policy-making) that should be made in order to increase resilience of cereals production under changing climatic conditions in the SSA region. I would particularly like to see a paragraph on how social policies to reduce vulnerabilities could play an important role in increasing resilience and adaptive capacity of the local populations.

Thank you. We have reworked the conclusions by adding more information on policy implications and future research that could enhance adaptation in southern Africa.

I also have a number of minor comments:

- L26: “growing” instead of “bourgeoning”.

Done

- L33: Please define WEF prior to the use of the acronym.

We have removed the sentence that used the acronym WEF and used it later in the manuscript in section 2 where it is fully explained.

- L38: “under changing climatic conditions” instead of “in an environment of climate change”.

Done

- L41: In which region of the world? Worldwide? Africa? Sub-Saharan Africa? Please clarify.

We have included southern Africa in the sentence

- L42: Is water deficits caused by competing sectors or by climate change (or both)? Please clarify.

Thank you. We have restructured the sentence as follows: “Already southern in southern Africa, climate change is evidently causing agriculture to struggle to meet the food demands of a growing population, mainly due to water deficits caused by increased demand from competing sectors as well as increased frequency and intensity of droughts”

- L45: Is rising sea level an important climatic determinant when it comes to food security? If so, please provide references for this statement (currently, ref number 9 is too broad to support this statement).

Yes, rising sea levels affects agricultural land particularly water managed areas. We have added two references

- L56: What is a “small” impact? Please clarify (I also suggest avoiding using such fuzzy words).

Thank you. The sentence has be restructured as “Although the impacts of climate change on food production are evident throughout the world, geographically they are unevenly distributed, with losses felt mostly in arid and semi-arid regions.”

- L61: “on water resources”.

Done

- L65: Any uncertainty range for these numbers?

We could not find any uncertainty range for the trends in temperature from the quoted literature.

- Figure 1: It would be great to add more information in this Figure, as a basemap. For instance, the different types of climatic zones or the different types of cereals productions.

We have added more information on Figure 1 by adding rainfall distribution in the region

- L117: That is instead of “that’s”.

Corrected

- L124-132: Is this paragraph really needed (or need to be that long)? This is quite technical and off-the topic. I would suggest reducing the length of this paragraph, while increasing the length of other paragraphs (based on a wider review of existing studies).

We consider this section important in that it notifies the reader of the main causes of the climate changes devastating the region. We have provided more statistical information in this section to further highlight the impacts of the ENSO. This then drives the reader to understand why the changes illustrated by Figures 3 and 4. In fact we made significant changes to the whole section for a smooth and better flow.

- L133: redundancy

Corrected by removing the repeated phrase

- L143: what are these changes? Please specify and elaborate on this.

We have highlighted the changes being caused by climate change in the water, energy, food, biodiversity and health sector in this section. Significant changes have been made in section 3

- L145: Please provide more references to support this crucial claim. The ref 42 is only about food security, and not about housing, employment, etc.

We added reference 18 where this is stated

- Figure 2: As said, it would be great to also have a figure showing future impacts of climate change (e.g. under different emissions scenarios) on temporal change in annual and seasonal rainfall in the SADC regions. This way, readers can easily grasp the extent of the impacts, compared to historical conditions.

We have added text to section 3 to further explain the future impacts of climate change on agriculture. Current impacts of climate change on agriculture is shown in Figure 5.

- L160: Please provide a scale of the CMI values, as many readers might not be aware of how the CMI is computed, what is its scale, and hence what -0.80 means (High? Medium? Low?).

We have provided the Vorosmarty CMI scale and explained and clarified the -0.80 classification of the region

- L163: what is huge? Please clarify (and I suggest avoiding the term ‘huge’ in any case, as it is highly subjective).

Thank you for observing this. We have re-written this paragraph providing statistics and numbers in order to be specific.

- Figure 3: since this is a review paper, I suggest including a figure from a different source, rather than using another figure from Nhamo et al.

We thought it would be appropriate to generate our own data to coincide with the period of crop yield data. Available datasets and maps are very old and do not reflect the current changes in aridity.

- L180-184: Please give some numbers and references to support these statements.

Thank you. We have provided numbers and further references

- L185: Please clarify what is the Mann-Kendall trend and what a value <0.01 means. A number of readers might be unaware of this trend index.

We have provided the definition of the Mann-Kendall trend to clarify the value <0.01

- L194: Why is that? Please explain how different South-African agriculture is.

We have added text to explain why South Africa contributes more to the regional food basket. The sentences now reads as follows: “Whilst 95% of the agriculture in the rest of other SADC countries is rainfed and vulnerable to climate change and variability, 30% of South Africa’s agriculture produce comes from irrigation and it exports part of the produce to some countries in the region [32].”

- Figure 4: what is ReSAKSS data? Please specify and define the acronym prior to its use.

We explained the meaning of ReSAKSS in section 1.1.

- Table 1: should be more discussed in the text.

We have added a paragraph discussing Table 1 in section 5

- L212: socioeconomic and climatic changes are two different things.

Thank you. Socio-economic has been removed

- L214: you could refer here to the new IPCC scenarios, i.e. socioeconomic scenarios (Shared socioeconomic pathways -- SSPs) and climate scenarios (Representative concentration pathways --RCPs).

We have also added more information on scenario planning including SSPs and RCPs in section 5 as highlighted in blue

- L215: please provide a few definitions for the terms that you use (reflexivity, responsiveness, and revitalization).

We have also defined the governance capabilities in the section 5 as highlighted in blue

- L235: At the end of this section, you could also briefly discuss a few innovative tools that can help identify appropriate adaptation options. One of this tool is the climate twins approach (Rohat et al. 2017 https://link.springer.com/article/10.1007/s11027-016-9708-x ; Rohat et al. 2018 https://www.emeraldinsight.com/doi/full/10.1108/IJCCSM-05-2017-0108), which has been applied in Sub-Saharan Africa to enhance resilience and food security at the local scale (see report from CGIAR https://ccafs.cgiar.org/publications/climate-analogues-finding-tomorrow%E2%80%99s-agriculture-today#.XCycCFxKiUk).

Thank you. We have added a paragraph on an innovative tool that can be used in adapting agriculture to climate change in the last paragraph of the section 5

- L247: Conclusion should not finish on such a negative side, but instead should reflect and discuss the way forward, and opportunities, and potential solutions to increase resilience.

We have also rewritten the conclusions as highlighted in blue.

Reviewer 2 Report

The paper provides an interesting work on reviewing the topic of crop production linked to climate change, and especially concerning the drought and water scarcity issue in Southern Africa countries. Authors give an overview of the water issue in relation with increasing irrigated area worsened by the aridity and the low capacity to reach food security; however, there are some comments and discussions to improve the study. Please refer to the comments below:

Comment 1: Authors should revise the structure of the entire manuscript in order to give a better understanding of the background of the topic, the objective of the review, and the review itself. The order of presenting the chapters can give a fluent reading to readers. I suggest revising the flow of the paper for a better understanding. Authors might summarize the different steps in the introduction section before entering in detail.

Comment 2: Line 19. Rather than “The review assessed trends…” is it better to say: “The review addressed trends…”?

Comment 3: In most part of the paper, the authors mentioned to technical terms as readers should know a priori. I suggest to write the acronyms in full (line 33; WEF might be Water-Energy-Food (WEF) Nexus?) the first time you mention. I also suggest to explicit the term “moisture” as I would refer to the “soil moisture” if this is the case. 

Comment 4: Line 23. “The region is described…”. Which region? Please, explicit the study area.

Comment 5: Line 114. What’s CAAPD?

Line 197, what’s ReSAKSS?

Comment 6: Line 160. Authors mentioned the CMI was assessed to be -0.80. I suggest to explain the methods deeper in detail or eventually to report the equation used and what sources authors used to calculate the index.

Comment 7: In chapter 4, authors express the relation between rainfall variability and crop production. I suggest using the Standardized Precipitation Index (SPI), which was developed to measure precipitation anomalies and characterize meteorological drought periods.

Comment 8: Please, explicit the data source for precipitation and crop production time-series in Figure 4.

Additional comment

I promote authors to provide a meaningful picture with a flowchart describing the interaction of forcing factors on water availability and food security. The picture aims to resume the review and provide the risk of climate change related to the water demand and crop production on the study area. Adaptation strategies might be added to show solutions to overcome the problem.

Author Response

Thank you for the time to review the manuscript. We have extensively revised the manuscript according to comments received, and changes are shown in blue.

Responses to comments are given in blue

The paper provides an interesting work on reviewing the topic of crop production linked to climate change, and especially concerning the drought and water scarcity issue in Southern Africa countries. Authors give an overview of the water issue in relation with increasing irrigated area worsened by the aridity and the low capacity to reach food security; however, there are some comments and discussions to improve the study. Please refer to the comments below:

Thank you for the valuable comments that we considered to improve this paper which we have extensively revised.

Comment 1: Authors should revise the structure of the entire manuscript in order to give a better understanding of the background of the topic, the objective of the review, and the review itself. The order of presenting the chapters can give a fluent reading to readers. I suggest revising the flow of the paper for a better understanding. Authors might summarize the different steps in the introduction section before entering in detail.

Thank you. We have added section 1.1 in which we describe the methodological framework. We have also provided Figure 2, which is a methodological flowchart of the study.

Comment 2: Line 19. Rather than “The review assessed trends…” is it better to say: “The review addressed trends…”?

Thank you. Done

Comment 3: In most part of the paper, the authors mentioned to technical terms as readers should know a priori. I suggest to write the acronyms in full (line 33; WEF might be Water-Energy-Food (WEF) Nexus?) the first time you mention. I also suggest to explicit the term “moisture” as I would refer to the “soil moisture” if this is the case.

Thank you. We have explained acronym (WEF) and clarified what we mean by moisture in the context of the paper (soil moisture and rainfall). This is now clarified in the first sentence of the introduction.

Comment 4: Line 23. “The region is described…”. Which region? Please, explicit the study area.

We replaced the word region with southern Africa.

Comment 5: Line 114. What’s CAAPD?

We have written the full name for CAADP ‘Comprehensive Africa Agriculture Development Programme

Line 197, what’s ReSAKSS?

We have fully given the name for ReSAKSS, ‘Regional Strategic Analysis and Knowledge Support System’

Comment 6: Line 160. Authors mentioned the CMI was assessed to be -0.80. I suggest to explain the methods deeper in detail or eventually to report the equation used and what sources authors used to calculate the index.

We have provided more information on how the CMI was calculated and provided the used equations in section 3

Comment 7: In chapter 4, authors express the relation between rainfall variability and crop production. I suggest using the Standardized Precipitation Index (SPI), which was developed to measure precipitation anomalies and characterize meteorological drought periods.

Thank you. Instead of using the SPI, we have added statistical information in the text to further aid the now Figure 5. We wanted to show the relationship in the trends of rainfall and cereal production over time without detailing much on drought. Added information is highlighted in blue. Using SPI would be emphasising and digging dipper on drought issues, which would divert us from the subject.

Comment 8: Please, explicit the data source for precipitation and crop production time-series in Figure 4.

We have provided the full source of data for now Figure 5

Additional comment

I promote authors to provide a meaningful picture with a flowchart describing the interaction of forcing factors on water availability and food security. The picture aims to resume the review and provide the risk of climate change related to the water demand and crop production on the study area. Adaptation strategies might be added to show solutions to overcome the problem.

Thank you. We have provided Figure 1 to clarify the interactions of the major points of the study.

Round 2

Reviewer 1 Report

Dear authors,

Please let me congratulate you for this revised manuscript, which I feel has largely improved compared to the previous version.

All of my comments have been addressed and I have no further major comments to make. I would simply like to ask for a complete spellcheck of the manuscript before its final acceptance and publication.